# A Controller for Robots to Autonomously Control Fog Machine

Adrian Lozada, Villa Keth, Uthman Tijani, Micheal Klein, Zhao Han*

*RARE Lab, Department of Computer Science and Engineering, University of South Florida*, Tampa, Florida 33620, USA

*Correspondence email: zhaohan@usf.edu

*Abstract*—**Typical fog machines need manual activation and human monitoring. This creates a problem that robots cannot interface with those fog machines to autonomously controll it for potential augmented reality (AR) applications, e.g., augmented to a fog screen. To solve this issue, we replaced the fog machine's manual remote with a custom PCB containing an Arduino microcontroller, where we implemented a programming interface that can be used by ROS-enabled robots. Besides Arduino, it has a latching relay and a rectifier circuit. The latching relay effectively "presses the button" to emit fog, while the rectifier reads the machine's signals to detect when it is hot and ready to use. The electrical design carefully separates high-voltage lines from control signals, and a 3D-printed enclosure keeps everything safe and accessible. For the programmable interface, it allows ROS to control the fog machine seamlessly, letting the robot toggle the fog output automatically. As a result, researchers can quickly adapt an off-the-shelf fog machine for various human-robot interaction studies, especially in settings where traditional projection surfaces are unavailable. The code, 3D models, PCB files, and documentation are available on GitHub at https://bit.ly/4b1Mq8j.**

*Index Terms*—**Fog machine controller, augmented reality (AR), robot communication, human-robot interaction (HRI)**

## I. Introduction

Human-robot interaction (HRI) researchers have investigated a variety of communication modalities, including arm movement [1], gaze [2], and augmented reality (AR) [3]. Projector-based AR is appealing because it can overlay visual images onto the robot's environment in a way that multiple people can see without headsets or other specialized devices [4]. However, projector-based AR typically requires a suitable flat projection surface, which may not exist in unstructured environments, e.g., search and rescue or construction sites.

To address the lack of flat surfaces, we created a portable midair fog screen [5] that serves as a projection plane for AR content. By allowing a robot to generate its own projectable surface, this fog-based technique using fog machines can expand how HRI researchers use projected AR in settings where flat surfaces are absent or obstructed.

However, although commercial off-the-shelf fog machines can produce suitably dense fog for mid-air projection, these machines are typically designed for manual activation—requiring a person to hold a button—and they only emit fog after reaching a certain operating temperature. For autonomous HRI scenarios, the robot itself must (1) power and activate the fog machine, (2) monitor its readiness status, and (3) integrate seamlessly with standard robotics software, Robot

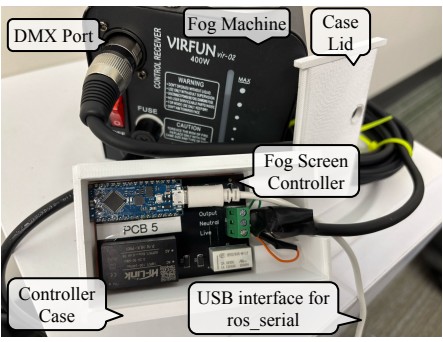

Figure 1. Our custom fog machine controller circuit on a PCB board allows autonomous control by ROS-powered robots through ros_serial.

Operating System, ROS). Those are the enabling technological contributions of this work, with a PCB board (Fig. 1 bottom) available to the VAM-HRI community.

In our prior work on a fog-screen-robot system [5], we only briefly described our custom fog machine controller about one column in an eight-page paper. This left out details like the full electrical schematic, component choices, and software integration under-explored. Consequently, it lacks a thorough guide for replication. In this paper, we present the complete hardware design and further detail the ROS-based software implementation. We discuss how we tackled technical issues to create a custom fog machine controller that plugs directly into a standard DMX port, eliminating the need to modify the fog machine's internal circuitry.

We started by reverse-engineering the original handheld controller to understand which wires were toggling the fog output and which signals indicated "ready to produce fog". From there, we designed a circuit board to integrate a microcontroller and built and tested the circuit as seen in Fig. 2. After confirming the circuit's functionality, we built and tested printed circuit boards (PCBs) with an Arduino microcontroller, a relay, and a rectifier circuit as in Fig. 1.

The objective is to give other researchers a straightforward, open-hardware and open-software solution for adding a fog machine to their own robotic projects. Our design is adaptable for all brands of fog machines with a similar DMX port. Our code and custom controller allow for autonomous communication between the fog machine and a ROS-powered robot. Autonomous control of fog output opens up further avenues to adapting this system to new applications, e.g., forming a

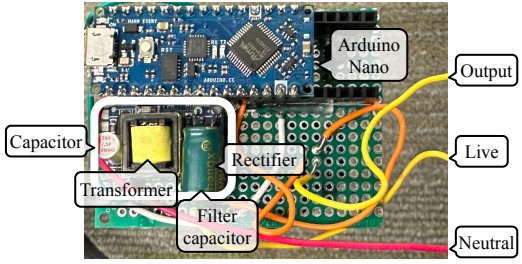

Figure 2. To test our custom controller design, we first built a prototype circuit. Key components include an Arduino Nano Every microcontroller, a relay, rectifier, transistors, resistors, and diodes. The Arduino turns the fog machine on or off via a relay below the Arduino. The diodes are on the other side of the circuit board.

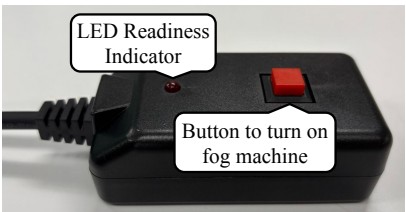

Figure 3. The handheld fog machine controller has an LED to indicate the machine's readiness for producing fog and a red button to operate the fog machine, preventing autonomous fog production.

fog screen that the robot can autonomously project onto [5].

## II. HARDWARE

### A. Reverse-Engineering Original Manual Controller

Commercial off-the-shelf fog machine [6, 7, 8, 9, 10, 11] comes only with a physically operated controller (Fig. 3) that requires manually pressing a button to turn the machine on or off. The red LED indicates when the machine is ready to produce fog. The controller is connected to the fog machine via a DMX cable port [12], which, as seen in Fig. 4, has three connections that we labeled: neutral, live, and output. The neutral and live wires carry 120VAC.

The problem with the original controller is that it cannot communicate with or control via a ROS-enabled robot [13] because it requires manual activation to control the machine, making it unsuitable for autonomous use. Additionally, the manual controller uses 120VAC voltage, which cannot be connected directly to a robot or a microcontroller, e.g., an Arduino, that connects to the robot.

To build the custom controller, we needed to understand how the original controller inside the manual control worked.

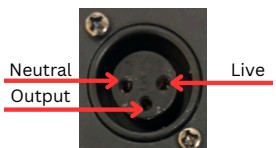

Figure 4. DMX port with the output, neutral, and live terminals. The manual control and our Arduino-based controller use it to control the fog machine, i.e., turn it on/off and read whether the machine is ready to produce fog.

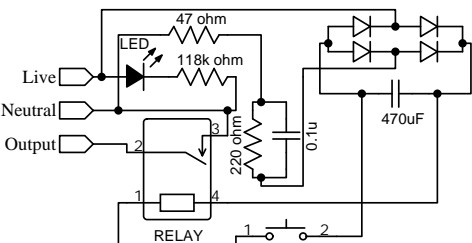

Figure 5. Schematic we created for the original fog machine controller, including three resistors (47 Ω, 118 kΩ, and 220 Ω), a 470 μF capacitor, a relay, and a bridge rectifier. It is designed to control the power supply, with connections for live and neutral inputs and output to the fog machine.

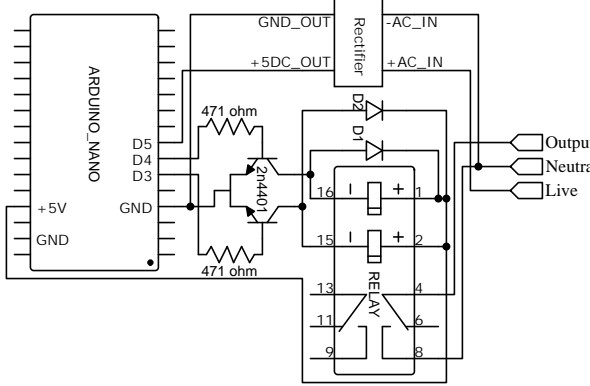

Figure 6. Schematic of our custom fog machine controller circuit: Arduino Nano Every microcontroller for automation, relay for switching the fog machine on and off, and the rectifier for determining the fog machine's readiness.

So, we opened the controller and sketched the connections and components. Then, we created a schematic of the original controller as shown in Fig. 5. In summary, when the push button is pressed, the controller connects the output and neutral wires using a relay. The red LED lights up when the fog machine detects that it has reached the appropriate operating temperature.

### B. Building Custom Controller for Robot Autonomy

Our design objectives are: (1) Safely interfacing with the high-voltage lines of the fog machine, (2) automatically detecting when the fog machine had reached the appropriate temperature to emit fog, and (3) allowing the robot to control fog output using standard robotics software, i.e., ROS.

We achieved these objectives by wiring a latching relay that performs the "pressing the button" task instead of the human user. We also used a full-bridge rectifier to determine the readiness of the fog machine. Figure 2 shows our first prototype on perfboard, while Figure 1 shows our finalized PCB version in a 3D-printed enclosure with Figure 7 being a closed-up. The following sections provide more details about the design of the custom controller.

*1) Latching Relay Functionality:* To safely switch the fog machine on and off, we used a relay to connect the output

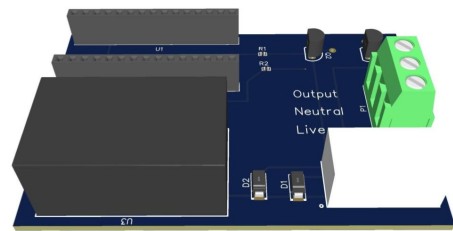

Figure 7. 3D view of the fog machine controller PCB.

wire to the neutral wire. Specifically, we used a HFD2/005-S-L2 latching relay [14], which has two coils: the set and reset.

The set coil, when energized, closes the relay contacts and sets the relay to the on state. On the other hand, the reset coil, when energized, opens the relay contacts, setting the relay to the off state. This dual-coil mechanism conserves energy since we only need to power the relay's coils when changing its state. In our design, the relay's contact is connected between the output and neutral wires. Energizing the set coil connects the output to neutral, activating the fog machine.

*2) Transistor Switches for Relay Control:* Because the relay requires about 40mA to actuate [14], it exceeds the Arduino Nano Every's per-pin current limit of 15mA at 5V [15]. To prevent overcurrent, we placed two 2N4401 NPN transistors [16] between the Arduino pins and the relay's set/reset coils.

Transistors are semiconductor switches with three terminals: collector (C), base (B), and emitter (E). A small current into the base allows a larger current to flow from the collector to the emitter. In our design, each transistor's collector is connected to the negative side of one relay coil, and its emitter is tied to ground. The "set" coil is driven by Arduino pin D4 through one transistor, and the "reset" coil is driven by pin D3 through the other.

To limit the base current, each transistor has a series resistor of $471\,\Omega$. The calculation for this resistor value, along with the detailed derivation using Ohm's law and the transistor's base-emitter drop, is provided in the Appendix. This setup ensures that only about 9mA flows from each Arduino output pin, within the board's safe operating range.

*3) Flyback Diodes for Inductive Load Protection:* Relay coils are inductive loads that store energy in the form of a magnetic field while current is flowing through them. When the current is switched off, the energy stored in the magnetic field must dissipate, if not, causing a high voltage spike, also known as back EMF (Electromagnetic Field). This voltage spike can damage components in the circuits.

To protect the transistor from the back EMF, we placed flyback diodes across each relay coil. These diodes provide a path for the current to dissipate and prevent the current from passing through the transistors. The flyback diodes are connected in reverse bias across each relay coil, which means that the anodes are connected to the negative (ground) side of the coils and the cathodes are connected to the positive sides. Under normal conditions, when the transistor allows current to flow through the relay coil, the diode does not conduct

because it is in reverse bias. However, when the transistor turns off and the magnetic field collapses, the diodes become forward-biased due to the induced back EMF, which dissipates the current.

*4) Readiness Signal Detection:* The fog machine indicates it is ready for use by setting the Live to 120VAC. However, we could not connect the Live wire directly to the Arduino Nano Every because it only operates with DC voltages, and the voltage has to be below 5VDC. To address this, we used a full-bridge rectifier circuit [17] with a built-in step-down transformer similar to the ones used in phone chargers, to convert AC to DC and lower the voltage from 120VAC to 5VDC.

The circuit steps down 120VAC to 5VAC using the step-down transformer, as seen in Fig. 2. Then, the full-bridge rectifier converts the alternating current to a direct current. However, due to the oscillating nature of the AC input, the resulting DC output is not perfectly smooth and contains voltage ripples corresponding to the AC frequency. To smooth out these voltage ripples, a capacitor is placed across the output of the rectifier. The capacitor acts as a filter by charging when the voltage rises and discharging when it falls, effectively leveling out fluctuations and providing a steadier DC voltage. We could then connect the output of the rectifier to one of the Arduino Nano Every's pins and check if it is HIGH (5V). If it is, it indicates that the fog machine is ready to produce fog.

This method is effective because we only need to detect whether the live wire is HIGH or LOW—a binary signal. The circuit modifies the voltage and current levels without altering the binary signal being sent. This allows us to reliably monitor the status of the fog machine while ensuring the safety of the microcontroller.

*5) Wiring:* We wired the custom controller by first connecting the output and neutral wires of the fog machine to the relay's input terminals.

Next, we connected the live and neutral wires of the fog machine to the full-bridge rectifier, which converts 120VAC to 5VDC. The negative output of the rectifier is connected to ground, whereas the 5VDC output of the rectifier is connected to pin D5 of the Arduino to determine whether the fog machine is ready to produce fog.

To control the relay, we connected Arduino pins D3 and D4 to the bases of two NPN transistors that toggle the relay coils. The emitters of both transistors are connected to ground, whereas the collectors are connected to the negative terminals of the relay coils.

To protect the Arduino and limit the current entering the transistor bases, we added $471\,\Omega$ resistors between the Arduino pins (D4 and D3) and the bases of the NPN transistors.

Additionally, flyback diodes are connected across the relay coils to protect the transistors from voltage spikes (back EMF) generated when the relay is de-energized. The anodes of the diodes are connected to the negative side of the relay coils, while the cathodes are connected to the positive side.

*6) PCB and Enclosure:* We designed the PCB for ease of assembly and maintenance (Fig. 1 & 7). It features a

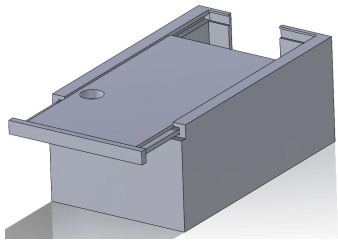

Figure 8. To protect our custom PCB and people from the high-votage, we designed a case with a slidable lid and 3D printed the case using PLA.

female header pin arrangement, allowing the Arduino Nano Every to be installed without soldering, making assembly and replacement more efficient. In addition, the PCB includes a three-input screw terminal for easy connection to the fog machine. To assemble the controller, simply insert the Arduino Nano Every–ensuring its USB port faces to the right, as shown in Fig. 1—into the headers. To connect the DMX cable's wires, identify each one using Fig. 4, then tighten each into the screw terminal with a small flat-head screwdriver.

To protect the electronics and keep the high-voltage lines inaccessible, we 3D-printed a custom enclosure (Fig. 8). The box dimensions accommodate the PCB (approximately 46mm height, 65mm in width, and 109mm in length) with a small margin, and it includes a sliding lid for quick access to the circuit board, rectangular openings (right) for the Arduino USB cable and DMX port wires, and small holes (top) aligned with status LEDs.

## III. SOFTWARE AND USAGE NOTES

Our software provides an interface between the Arduino-based fog machine controller and the robot through ROS. This communication relies on the *rosserial* library [18], which establishes a serial connection over USB. In practice, the Arduino advertises two ROS services named `/fog_machine/turn_on` and `/fog_machine/turn_off`. These calls let the robot activate or deactivate the latching relay connected to the fog machine. Before powering on the machine, the Arduino checks whether the rectifier signal indicates that the device is heated and ready to produce fog. If the rectifier's output is not high, the Arduino delays the activation until the machine finishes warming up, thus ensuring that any subsequent projection task can begin immediately.

The Arduino code is found in the file `ros_arduino_service.ino`. It initializes the pins for the relay coils and rectifier input and sets up service callback functions to handle the on/off requests. Within these callbacks, the software blocks until the rectifier signal indicates readiness before triggering the relay coil to supply power to the fog machine. Once the relay has been briefly energized, the Arduino releases it to conserve energy, since a latching relay only needs current when changing states.

On the robot side, it is necessary to run the node that connects the Arduino to the rest of the system. For example, on Ubuntu 18.04 with ROS Melodic installed, the three core *rosserial* packages can be installed by issuing commands for `ros-melodic-rosserial`, `ros-melodic-rosserial-arduino`, and `ros-melodic-rosserial-python`. After uploading the Arduino sketch, the user starts the serial node by invoking the corresponding script in the *rosserial_python* package, pointing it to the correct port for the Arduino (often found under `/dev/tty<port name>`).

Any other ROS node can invoke the fog machine services simply by calling `/fog_machine/turn_on` or `/fog_machine/turn_off`. Developers can write a service client in C++ or Python by following the standard ROS tutorials. When `/fog_machine/turn_on` is called, the service callback checks if the device is fully warmed, activating the relay only when the rectifier input is confirmed high. If users want additional control logic, such as partial fog output or varying fog pulse durations, they can easily modify the Arduino service callbacks to perform more complex actions.

Before running the software, it is important to ensure that all PCB connections have been connected correctly and that the board is safely enclosed to isolate any high-voltage wiring. Once the hardware is verified and the node is started, subsequent calls to `/fog_machine/turn_on` and `/fog_machine/turn_off` provide a straightforward way to integrate autonomous fog deployment into a robot's communication flow.

## IV. CONCLUSION

In this paper, we demonstrate how an off-the-shelf fog machine can be transformed into an autonomous device for robotic applications using our custom low-cost add-on controller. By reverse-engineering the fog machine's original manual remote and safely connecting to its high-voltage lines with a relay and rectifier, we designed a custom PCB from a prototype perfboard. Also contributing to a ROS interface, our work enables commonly ROS-powered robots to detect when the fog machine is heated and to autonomously produce fog output as needed—without requiring a human operator.

In summary, our controller utilizes the fog machine's DMX port, avoiding any invasive modifications to its internal electronics and preserving compatibility with various fog machine models. We have open-sourced both the hardware and software so that researchers can replicate or adapt our design. The Arduino code allows users to easily adjust pulse patterns or intervals to manage the amount of fog emitted. Editing a few lines of code makes it possible to create intermittent bursts of fog with adjustable density and frequency, opening up new avenues for VAM-HRI systems and studies.

To conclude, no work is withoug limitations. One current limitation of our controller is that it cannot detect if the fog machine itself has been powered off at the main supply. It only monitors the machine's readiness via a heating indicator line. Future iterations may address this issue by incorporating another rectifier into the circuit or using computer vision to detect whether the fog machine is on or off.

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

APPENDIX

In this appendix, we provide a step-by-step derivation of the resistor values used for the transistor base drive.

The base current $I_\text{B}$ is determined by Ohm's law:

$$I_\text{B} = \frac{V_\text{Arduino} - V_\text{BE}}{R_\text{B}}$$

where:

- $V_\text{Arduino}$ is the output voltage of the Arduino (typically 5 V),
- $V_\text{BE}$ is the base-emitter voltage drop (approximately 0.7 V for a 2N4401 transistor), and
- $R_\text{B}$ is the resistor connected between the Arduino output and the base.

For our design, we selected $R_\text{B} = 471\,\Omega$. Substituting the values, we have:

$$I_\text{B} = \frac{5\,\text{V} - 0.7\,\text{V}}{471\,\Omega} \approx \frac{4.3\,\text{V}}{471\,\Omega} \approx 9.13\,\text{mA}.$$

This calculated base current is within the safe operating limits for the Arduino's digital pin, ensuring that the transistor saturates properly without overloading the microcontroller.