# OpenReview forum: "A Controller for Robots to Autonomously Control Fog Machine"
_humanrobotinteraction.org/HRI/2025/Workshop/VAM — HRI 2025 Workshop VAM Submission_

### Official Review · Reviewer_HiQ1 · 2025-02-28

**Rating:** 7
**Confidence:** 5

**Review:**

Paper Information

- Title: A Controller for Robots to Autonomously Control Fog Machine

- Submission Track: Early Deadline

- Page Length: 6 pages including references

Overall Recommendation

Accept

Executive Summary

This paper presents a well-documented hardware and software solution for enabling robots to autonomously control fog machines for AR applications. The work directly addresses a practical challenge in VAM-HRI research by creating an open-source interface between ROS-enabled robots and commercial fog machines, enabling new possibilities for projected AR in unstructured environments.

Relevance to VAM-HRI Themes

Workshop Theme Alignment

- Main theme: AR/VR for robot testing and diagnostics

- Justification: The paper creates fundamental hardware infrastructure for AR-based robot interactions, specifically enabling projected AR in environments lacking suitable projection surfaces. This work directly supports the workshop's focus on expanding AR capabilities in HRI settings.


Secondary Themes

- AR-based intent communication

- Architectures for AR/VR-based HRI

The paper successfully integrates these themes by providing a technical foundation that enables various AR-based interaction scenarios.

Technical Merit

Strengths

- Comprehensive documentation of both hardware and software implementation

- Clear circuit design with safety considerations for high-voltage components

- Practical integration with ROS through well-designed service interfaces

- Thorough explanation of component selection and design decisions

- Open-source release of all necessary files for replication


Some Areas for Improvement

- The current design cannot detect if the fog machine is powered off at the main supply

- Limited discussion of the testing methodology used to validate the controller

- Could benefit from more quantitative performance metrics

- Safety certification considerations could be more explicitly addressed


Methodology Assessment

- Well-structured engineering solution with clear motivation

- Strong attention to electrical safety and component selection

- Implementation details are clear, though systematic testing results could be expanded

Presentation Quality

- Clear progression from problem statement through implementation

- Technical writing is precise and accessible

- Excellent use of figures and schematics

- Good diagrams that effectively illustrate the hardware design

Impact and Innovation

- Addresses an unmet need in VAM-HRI research infrastructure

- Enables new research directions in projected AR for HRI

- Immediate utility for researchers working with projected AR in robotics


Suggested Improvements

1. Add quantitative performance metrics (e.g., response time, reliability statistics)

2. Include more details about testing methodology

3. Expand discussion of safety certifications needed for deployment

4. Consider adding error handling documentation

---

### Decision · Program_Chairs · 2025-02-26

Accept